# Capacity Analysis of Incentive Schemes in Opportunistic Networks

Ruoyu Feng , Shengming Jiang * and Zhichao Zheng

College of Information Engineering, Shanghai Maritime University, Shanghai 201306, China
* Correspondence: smjiang@shmtu.edu.cn

**Abstract:** Many incentive schemes address the selfishness issue in opportunistic networks and show performance improvement by simulations. However, the insights of incentive schemes that affect network performance are not clear. Network capacity analysis can reveal how factors affect performance, which is a guideline for new designs. To analyze incentive schemes, a well-defined mathematical model is necessary, which cannot be achieved by existing analytical models based on empirical formulas or types of incentive schemes. First, this paper proposes such a model to show the incentive degree with the incentive scheme, cooperation degree, energy usage, buffer usage, and security based on a quantum game model. Verification compares the model with delivery ratios that show impacts on selfish nodes in simulations under two typical incentive schemes. Then, network capacity is analyzed with this model and a sparse clustering regime that has similar mobility to opportunistic networks in order to show factors for future designs.

**Keywords:** opportunistic network; analytical model; incentive scheme; network capacity

## 1. Introduction

Mobility affects network capacity, which refers to the maximum achievable throughput due to the dynamic cooperation of nodes. Mobile models are discussed often [1]. For instance, sparse clustering regimes show that grouping nodes can increase the cooperation degree, and trading throughput for delay in clusters does not decrease network capacity as a whole [2]. While these studies usually assume nodes are rational, the cooperation degree may decrease due to selfish nodes, especially in opportunistic networks where end-to-end paths between sources and destinations change over time and are occasionally even disrupted [3]. Incentive schemes force nodes to forward messages in order to avoid long waits for the next opportunity, which increases co-operation as a way to prompt delivery ratios and decrease end-to-end contact times. To design more efficient incentive schemes, it is important to discuss factors affecting performance. This paper analyzes network capacity, with incentive schemes as the goal. In addition, our study can be applied to filter candidates for schemes that raise prediction-based, hybrid, or other networks with opportunistic networks such as opportunistic routing based on prediction [4].

To analyze network capacity, a mathematical model of incentive schemes is necessary for sufficient insight. Existing incentive schemes only show improvement by simulations, and existing analyatical models are not detailed enough. For example, a framework for multiple-criteria decision-making problem to rank protocol performance [5] lacks numerical formulas involving parameters. Game theory has been used to state nodes' strategies to analyze network traffic in transmission control protocol (TCP) with selfish nodes and peer-to-peer (P2P) networks with free-riding problem [6,7], and several analytical models for different categories based on classical game theory have been proposed [8–10] which do not find common characters of incentive schemes. A generic model can be built with quantum game theory [11], as quantum game theory extends the strategy space to search for optimal strategies in a wider range and uses the concept of entanglement to depict the

complex interaction among nodes accurately in the decision-making process. A model with these features can evaluate most incentive schemes and reveal their common character. However, the above model uses an empirical utility formula and focuses on numerical analysis, which simply places datasets into the model without other comparisons. In other words, the model does not have enough factors to work as a guideline to design new incentive schemes.

A well-defined formula should consider the characteristics of incentive schemes as well as other factors that affect cooperation. A node usually spends payoff, such as virtual money, reputation, exchanging messages, or something else, to stimulate others to forwarding, which can obtain the necessary currency. Cost and reward can represent an incentive scheme in a mathematical model with other properties of nodes. Several incentive schemes take energy usage and buffer usage into account. Therefore, this paper proposes an analytical model based on quantum game theory considering the energy usage and buffer usage, reward, cost, and cooperation degree, which shows the incentive degree and compares incentive degrees with the delivery ratio by simulations.

Network capacity analysis using the proposed model shows the impact of incentive schemes on network performance. During analysis, a mobile model in opportunistic networks is needed as well. The sparse clustering regime is similar to the mobility of opportunistic networks with nodes that work in groups with high cooperation possibility and clusters that move randomly where communications in groups are stable and movement is otherwise needed for opportunities. Furthermore, the scheduling policy is similar to the scenario in [4] for nautical wireless ad hoc networks. Therefore, the analytical model for incentive schemes is applied to a sparse clustering regime to analyze network capacity and discuss various factors.

As a summary, the contributions of this paper are as follows: (1) An analytical model based on quantum game theory considering the energy usage, buffer usage, reward, cost and cooperation degree is proposed. It is verified by a comparison between the incentive degree and delivery ratios according to simulations. (2) Network capacity is analyzed with the above model to show how incentive schemes affect network performance. The factors can be used as a guideline for designing efficient incentive schemes by considering these relations.

The rest of this paper is organized as follows. The mathematical model is detailed in Section 2. Verification with simulations is described in Section 3. Network capacity is analyzed with the model and sparse clustering regime in Section 4. The factors are discussed in Section 5. Finally, we conclude the paper in Section 6.

## 2. Mathematical Model for Incentive Schemes

### 2.1. Typical Incentive Schemes

Before detailing the mathematical model, we briefly touch on typical incentive schemes to see what can be taken into account. Most existing incentive schemes are reputation-based or credit-based [10]. They use currency-like metrics to reveal the participation degree for nodes forwarding messages [12].

In credit-based schemes, relay nodes can obtain credits by forwarding messages, then send their own messages with these additional credits. The better design and calculation of reward and price remains a challenge. Payoff according to the time-to-live of a message has been proposed in [12], which is direct to price difference. A Markov chain can be used to calculate price according to the society degree, selfish degree, and cooperation degree with buffer, energy, and interaction to stimulate nodes with high possibility [13]. The clearance credit center rewards relay nodes with extra credits to avoid nodes being unable to obtain enough credits to send messages due to selfish nodes who hold most credits and are not willing to forward messages. Therefore, attackers are not considered in our model or in other similar proposals, such as [8,10].

Other incentive schemes, such as reputation-based ones, usually replace credits with internal bounds for the same purpose. Punishment is used to restrain free-riding nodes

in peer-to-peer (P2P) networks via game theory [7]. Grouping nodes by bilinear pairing and cooperation probability prediction has been proposed in [14]. Evolutionary game theory replaces the classical theory for more precise cooperation evaluation [15]. With the exception of node behaviour under different policies, these bounds, such as cooperation probability, are similar to credits both in usage and principle. Thanks to the extended strategy space in quantum game theory, it is possible to simply consider increasing bounds as rewards and decreasing bounds as costs.

An environment involving movement path, link quality, and others can affect communications as a main domain. However, these do not affect incentive schemes directly, which is shown simply by a scenario argument without detailed considerations in the following analysis.

To sum up, for an analytical model, the cost and reward function can represent an incentive scheme. Other parameters such as buffer and energy can be considered as a part of cost, as they may reduce the cooperation degree. A simple argument briefly summarizes the impact of the environment. Furthermore, interactions among nodes must be considered, which can be solved by a quantum game model and Nash equilibrium.

*2.2. Quantum Game Model*

The proposed model has two parts, as shown in Figure 1. The quantum game model depicts how an incentive scheme motivates cooperation, and the utility formula $\$_m$ describes how a node is served. The utility formula $\$_m$, including the energy usage, buffer usage, and degree of security, accepts an incentive policy IP and entanglement $\lambda$ as arguments. In both the quantum game model and the utility formula, $\lambda$ communicates with a quantum game model to describe incentive processes. Therefore, the optimal value $\$^*$ is computed by the quantum game model with Nash equilibrium.

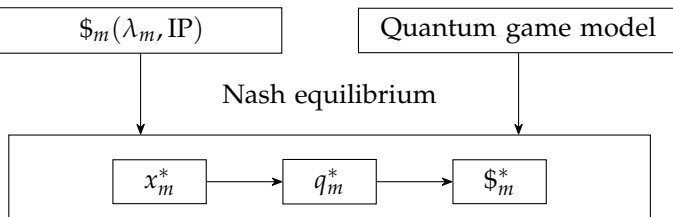

**Figure 1.** Model construction.

Quantum game theory is more suitable for analyzing incentive schemes because of the concept of entanglement and the extended strategy space, as discussed in Section 1. First, we briefly describe quantum game theory based on [11], shown in Figure 2. The actions of players in a quantum game ae based on a specific set of strategies. The state of node $m$ is represented by $|vac_m\rangle$, a state vector in a Hilbert space. The whole system state is $|H\rangle = |H_1\rangle \otimes |H_2\rangle \cdots \otimes |H_n\rangle$.

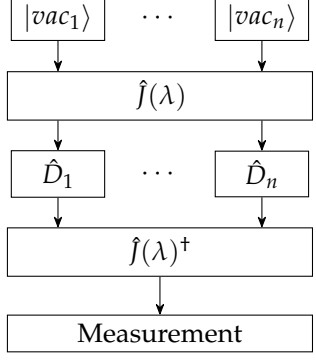

**Figure 2.** Quantum game model.

If players are dependent, entanglement must be used to express the system state in the above product. Entangling gate $\hat{J}(\lambda)$ with operation parameter $x_m$ defines how to combine the strategies of each node. This expresses how players affect others, such as encounters due to moving paths and selfish policies, especially in opportunistic networks; in our study, entanglement is considered as the cooperation degree. The model includes the following three stages:

1. The initial quantum state of the system is $\psi_{init} = \hat{J}(\lambda)|vac\rangle$, achieved by entangling all nodes togather.
2. Node $m$ operates on its state using unitary operator $\hat{D}_m$ with the feature $\hat{D}_m\hat{D}_m^{\dagger} = 1$.
3. With disentangling gate $\hat{J}(\lambda)^{\dagger}$ on the system, the final state is $\left|\psi_{fin}\right\rangle = |J\rangle(\lambda)^{\dagger}(\hat{D}_1 \otimes \hat{D}_2 \cdots \hat{D}_n)\hat{J}(\lambda)|vac\rangle$.

Each definition and proof are detailed in [11]. After forwarding $\left|\psi_{fin}\right\rangle$ to measurement, node $m$ can quantize the relayed data to others as follows:

$$q_m = \frac{x_m}{n}\left[e^{(n-1)\lambda} + (n-1)e^{-\lambda}\right] + \sum_{j=1,j\neq m}^{n}\frac{x_j}{n}\left[e^{(n-1)\lambda} - e^{-\lambda}\right]. \tag{1}$$

The optimal value $x_m^*$ can be computed by a utility formula with the Nash equilibrium for impacts on nodes. The Nash equilibrium is defined as follows:

$$\begin{cases} \dfrac{\partial\$_1}{\partial x_1} = \dfrac{\partial\$_2}{\partial x_2} = \cdots = \dfrac{\partial\$_n}{\partial x_n} = 0 \\ \dfrac{\partial^2\$_m}{\partial x_m^2} < 0\,, m \in \{1,2,\cdots,n\}. \end{cases} \tag{2}$$

Then, the optimal strategy $q_m^*$ and optimal utility $\$_m^*$ can be calculated with $x_m^*$ according to (1) and (3). The calculation depends on the utility formula, with the input incentive scheme to be discussed later. Then, we obtain a mathematical equation between an incentive scheme and $\$^*$, which can be used for evaluatation and reveal the insight if the utility formula depicts factors correctly. Moreover, $\$^*$ is called the incentive degree in this study.

### 2.3. Utility Formula

The utility formula describes the incentive degree of a node with entanglement $\lambda_m$, reward function $r(q_m)$, cost function $c(q_m)$, and constant cost $c'$, as follows:

$$\$_m = \lambda_m q_m - r(q_m) - c(q_m) - c'\,, \tag{3}$$

where $r(q_m)$ is represented as the incentive scheme. According to the introduction of incentive schemes in Section 2.1, an incentive scheme consists of a reward function $R(q_m)$ and a price function $P(q_m)$. The incentive degree is more dependent on their relation than each individual calculation, and the calculation makes the incentive process more rational and balanced. Therefore, in the utility formula, only the relationship between $R(q_m)$ and $P(q_m)$ is considered, as follows:

$$r(q_m) = q_m\frac{R(q_m)}{P(q_m)}. \tag{4}$$

We show the correction with simulations in Section 3. Entanglement $\lambda_m$ means the cooperation degree determined by the scenario, such as selfish degrees, social groups, movement paths, etc., which can be avoided by fixing the entanglement in the model during analyses.

The cost constant $c'$ is for each transmission, such as the fixed energy consumption [16]. The cost function $c(q_m)$ describes what relay nodes have to pay for when transmissions

succeed, which is made up of an energy cost $E(q_m)$, buffer cost $B(q_m)$, and security cost $S(q_m)$, as follows:

$$c(q_m) = E(q_m) + B(q_m) + S(q_m) . \tag{5}$$

The security cost is simple:

$$S(q_m) = sq_m^2 , \tag{6}$$

where $s$ is the willingness to forward messages, determined by reasons such as security. Unlike entanglement $\lambda$, $s$ is a node's own attribute and is not affected by other nodes.

The incentive degree does not continue to increase with larger buffer size and energy capacity, as it is limited by the cooperation between nodes. For instance, a node can only forward messages to parts of others, even if it has infinite buffer and energy. Therefore, they should have asymptotic values. Furthermore, the slope change for the buffer is slower than that for energy, as the energy capacity grants a node the basic ability to participate in forwarding, instead of enough resources, such as buffer and the cooperation degree. To sum up, they are defined as follows:

$$E(q_m) = \left( \frac{E_c}{E_s} \right)^5 r(q_m) \text{ and } B(q_m) = \left( \frac{B_c}{B_s} \right)^3 r(q_m) , \tag{7}$$

where $E_c$ is the energy cost per message, $E_s$ is the energy capacity, $B_c$ is the buffer cost per message, and $B_s$ is the buffer size.

### 2.4. Model With Incentive Scheme

To obtain a model for a special incentive scheme, $P(q_m)$ and $R(q_m)$ should be defined in the utility formula, and the Nash equilibrium then applied to the qunatum game model with the utility formula.

Most incentive schemes are made up of a price function $Z$ and reward function $R$. The two functions are usually independent on $q_m$ and $\lambda_m$, especially in credit-based schemes. In these cases, $P(q_m)$ and $R(q_m)$ are linear functions of $q_m$, which can be defined as

$$P(q_m) = Zq_m \text{ and } R(q_m) = Rq_m . \tag{8}$$

Now, we can calculate the analytical model. According to (8), the utility formula is expanded as follows:

$$\$_m = \lambda_m q_m - \frac{R}{Z} q_m - \left( \frac{E_s}{E_s} \right)^5 \frac{R}{Z} q_m - \left( \frac{B_c}{B_s} \right)^3 \frac{R}{Z} q_m - sq_m^2 - c' . \tag{9}$$

Then, we apply (2) to obtain

$$\begin{aligned}
\frac{\partial \$_m}{\partial x_m} &= \frac{\partial \$_1}{\partial x_1} = \frac{\partial \$_2}{\partial x_2} = \cdots = \frac{\partial \$_n}{\partial x_n} \\
&= \frac{\partial q_m}{\partial x_m} \left\{ \lambda - 2sq_m - \frac{R}{Z} \left[ \left( \frac{E_c}{E_s} \right)^5 + \left( \frac{B_c}{B_s} \right)^3 + 1 \right] \right\} \\
&= 0 ,
\end{aligned} \tag{10}$$

where

$$\frac{\partial q_m}{\partial x_m} = \frac{e^{(n-1)\lambda} + (n-1)e^{-\lambda}}{n} . \tag{11}$$

Because optimal strategies are the same under Nash equilibrium, represented as $x_1^* = x_2^* = \cdots = x_n^*$, we can obtaun $x^*$ with the above results as follows:

$$\frac{\partial \$_m}{\partial x_m} = \lambda - 2sx^* e^{(n-1)\lambda} - \left[ \left( \frac{E_c}{E_s} \right)^5 + \left( \frac{B_c}{B_s} \right)^3 + 1 \right]$$
$$= 0 \tag{12}$$

and

$$x^* = \frac{\lambda - \left[ \left( \frac{E_c}{E_s} \right)^5 + \left( \frac{B_c}{B_s} \right)^3 + 1 \right] \frac{R}{Z}}{2se^{(n-1)\lambda}} . \tag{13}$$

Substituting $x^*$ in $q^*$ yields

$$q^* = x^* e^{(n-1)\lambda}$$
$$= \frac{\lambda - \frac{R}{Z}\frac{E_c}{E_s} - \frac{R}{Z}\frac{B_c}{B_s} - \frac{R}{Z}}{2s} . \tag{14}$$

Then, $\$^*$ can be computed by (3), which shows the incentive degree's relation with a special incentive scheme.

## 3. Verification of the Model with Simulations

The verification compares incentive degrees and delivery ratios with two typical incentive schemes, namely, the incentive and privacy-aware data dissemination scheme (IPAD) [12] and the probabilistic routing scheme based on game theory (PRGT) [13]. The two credit-based schemes use different calculation parameters. IPAD uses time-to-live for prices and degree of selfishness for rewards. PRGT evalutes social degrees and transmission history with a Markov chain to calculate prices and message importance for rewards. The following simulations are conducted by OMNet++ (https://omnetpp.org (accessed on 20 August 2022)) with the OPS framework (https://github.com/ComNets-Bremen/OPS (accessed on 20 August 2022)). The parameter setting is listed in Table 1 for maritime environments where communications are over long ranges, which is associated with sparse density of node points. Random movement is selected because it is the part of mobile model in the latter network capacity analysis used to show the average theoretically. Nodes' selfish behavior is partly cooperative [17] in our simulations. Certain parameters, such as movement and the type of selfish behavior, are reflected by $\lambda_m$ indirectly.

**Table 1.** Simulation parameters.

| Parameter | Value |
| --- | --- |
| Simulation time | 8 hours |
| Area size | $5 \times 5 \text{ km}^2$ |
| Transmission range | 300 m |
| Transmission speed | 1 Mbps |
| Node count | 100 |
| Message interval | 15 s |
| Message size | truncnormal $(1.5, 1)$ MiB |
| Buffer size | 56 MiB |
| Node selfish | bernoulli $(0.7)$ |
| Movement change interval | truncnormal $(30, 15)$ min |
| Movement angle delta | normal $(0, 30)°$ |
| Movement velocity | normal $(30, 10)$ mps |

Because successful transmissions in opportunistic networks are highly dependent on mobility and node forwarding, selfish nodes reduce delivery ratios [13] remarkably. Utility

$\$^*$ represents the incentive degree, which can be reflected by delivery ratios in the network with selfish nodes. In comparison, the co-domain of $\$^*$ is rescaled with the same co-domain of delivery ratios due to the different scalars. The rescaled values are analyzed by mean square errors (MSE), as shown in Table 2 and Figures 3 and 4. We analyze each of them in the following subsections. Another important metric for opportunistic networks is the contact time of the nodes. Because selfish nodes decrease contact times due to the lower forwarding rate and incentive schemes try to increase the rate shown by delivery ratios, it is implied in delivery ratios that there is a higher delivery ratio with a lower contact time if there are no extra factors, such as attackers.

**Table 2.** MSE between delivery ratios and $\$^*$.

| Parameter | IPAD | PRGT |
|---|---|---|
| scheme parameter | $5.18 \times 10^{-5}$ | $1.76 \times 10^{-4}$ |
| buffer size | $5.00 \times 10^{-4}$ | $7.04 \times 10^{-4}$ |
| energy capacity | $2.48 \times 10^{-5}$ | $4.14 \times 10^{-5}$ |

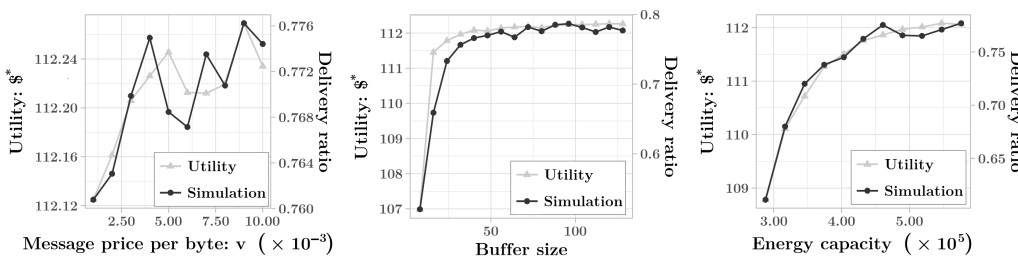

(**a**) comparison of $v$  (**b**) comparison of buffer size  (**c**) comparison of energy capacity

**Figure 3.** Verification with PRGT.

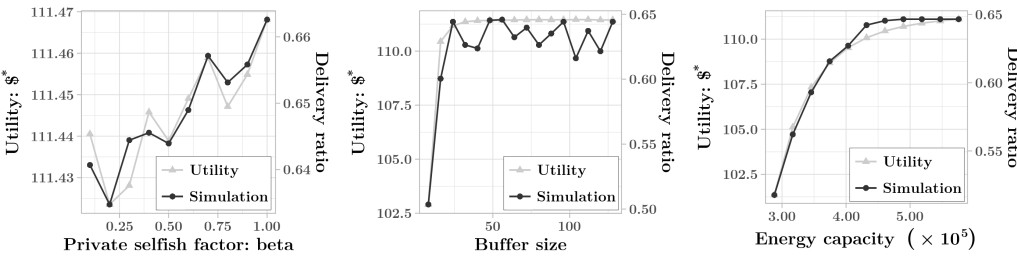

(**a**) comparison of $\beta$  (**b**) comparison of buffer size  (**c**) comparison of energy capacity

**Figure 4.** Verification with IPAD.

### 3.1. Incentive Schemes

The message price per byte $v$ in PRGT affects the message price and reward directly; thus, it is selected as an argument in this verification. For PRGT, $\$^*$ and delivery ratios from simulations are shown in Figure 3a.

In IPAD, the private selfish factor $\beta$ is the parameter for analysis, and is used in the payoff function for incentive. With a larger $\beta$, a larger price is needed and nodes have to forward more messages to obtain credit for their own messages, as shown in Figure 4a. As a whole, the similar relations show the correctness and generality of $r(q_m)$.

Furthermore, from $\$^*$ and delivery ratios, it can be found that the impact of parameters in incentive schemes is low. The main factor is the incentive strategy, represented by $r(q_m)$. The $\$^*$ of PRGT is larger than that of IPAD, which means that the degree of incentive of PRGT is greater than that of IPAD in the same scenario.

### 3.2. Buffer Size

Similar to the previous verification, delivery ratios and $*$ with buffer sizes are shown in Figures 3b and 4b. The two similar relations suggest that buffer sizes can affect performance, and the proposed model reflects the realistic relationship with buffer size. Intuitively, message sizes have similar results as buffer sizes, because they are inversely proportional in the model and work together to store messages in simulations.

An increasing buffer size means that a node can forward more messages or a larger message at a time, however, the increase is slowed down because of the restriction of nodes' movement paths. Therefore, too large a buffer size is useless for network performance unless used with a larger message size.

### 3.3. Energy Capacity

The selfish behavior caused by energy in simulations is that if the energy is lower than the safety value, forwarding is stopped unless nodes need credits. The energy capacity is set to simple values here instead of considering a real scenario for easier calculations and understanding. Delivery ratios and $*$ with energy capacities are shown in Figures 3c and 4c. These two similar relations suggest that this parameter can affect performance and the proposed model can reflect the realistic relationship.

Similar to messages size, the energy consumption per action has similar results to energy capacity, as they influence each other in the model and the simulations.

## 4. Capacity Analysis for Incentive Schemes

Network capacity can be analyzed with the proposed model by abstracting incentive schemes. For simplification, this analysis is based on an existing efficient mobile model of a sparse clustering regime [2], in which nodes' behaviour resembles the opportunistic networks with social groups discussed in Section 1. The network model is introduced at first. then the capacity and factors are discussed.

### 4.1. Traffic Model

We assume all nodes communicate with others at the same rate $\lambda$.

**Definition 1** (The throughput capacity of opportunistic networks)**.** *The throughput capacity $\lambda$ is that nodes can sustain $\Theta(f(n))$ bits per second if there are deterministic constants $c > 0$ and $c' < \infty$ such that*

$$\begin{cases} \lim_{n \to \infty} \Pr[\lambda(n) = cf(n) \text{ is feasible}] = 1 \\ \lim_{n \to \infty} \Pr[\lambda(n) = cf(n) \text{ is feasible}] < 1. \end{cases} \tag{15}$$

### 4.2. Network Topolopy

The sparse clustering regime considers $n$ nodes moving over a square with area $n$, and nodes are divided into $m = \Theta(n^v)(0 \leq v < 1)$ groups. Each group covers a disk with radius $R = \Theta(n^\beta)$. The average number of nodes per cluster is assumed as $q = n/m = \Theta(n^{1-v})$. When $v + 2\beta < 1$ (i.e., $mR^2 = o(n)$), the strong node cooperation in which communications are mainly dependent on mobility is considered as a sparse clustering regime.

The time is divided into slots. Cluster centers and nodes are independent and identically distributed (IID) and uniformly chosen among the whole network and their own cluster, respectively, at the beginning of each time slot. That is, each cluster center randomly selects a point in the network area as its new center and nodes randomly select positions in the contained cluster. Nodes transmit messages without movement.

### 4.3. Scheduling Policy

Nodes move in clusters and clusters move in the whole area such that different usage of high and low quality communications is exploited. The tradition models can be considered as all nodes under the same cluster with different node density. The scheduling policy defines how clusters work under the regime, which is shown in Figure 5.

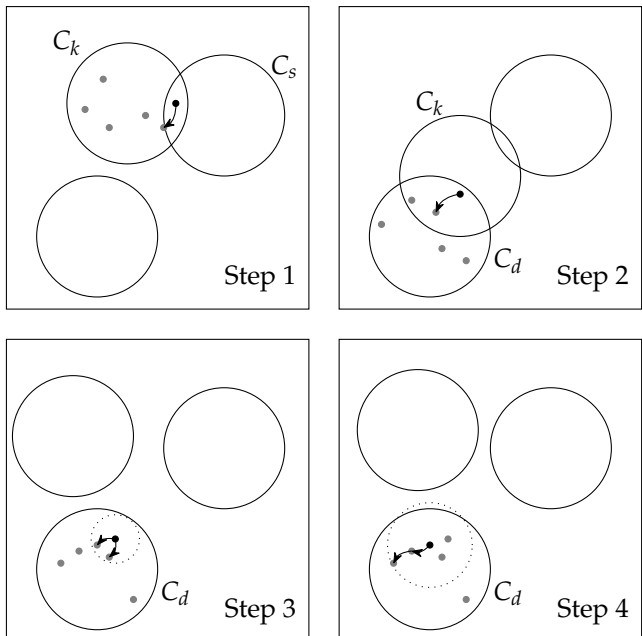

**Figure 5.** Scheduling policy.

1. A source creates relays $R_s$ in source cluster $C_s$ via multicast. The message is sent to a cluster $C_k(k = 1, \cdots, R_c{}^s)$ by one-hop unicast when the cluster meet others.
2. Relays $R_k{}^s$ are created in $C_k$ via broadcast. After clusters repeat these actions some number of times, the message reaches the destination cluster $C_d$.
3. Relays $R_d{}^s$ are created by broadcast in the destination cluster until the message is near to the destination node.
4. If the distance between the destination node and the current node is within a range $l^s$, the message is transmitted via $h^s$-hop unicast to reduce resource usage.

*4.4. Transmission Protocol*

If sources $i$ and $j$ want to send messages to destinations $k$ and $j$, respectively, for an arbitrary other node $k$, the following equation should be satisfied [18]:

$$|X_j - X_i| \geq \frac{\Delta}{2}(|X_i - X_k| + |X_j - X_l|),\tag{16}$$

where $X_i$ is the position of $i$ and $|X_i - X_j|$ is the distance between them. The basic inequation is often used in research directly, and is associated with incentive schemes similar to the physical model in [18].

To have transmissions, the incentive degree should have a lower bound, that is,

$$\$(\mu) \geq \$.\tag{17}$$

According to inequality properties, we have

$$\frac{\$_k}{\$_i + 1} \leq \frac{\$}{\$ + 1}.\tag{18}$$

If we assume successful transmission probabilities between nodes are the same, the closest node has the highest cooperation degree. Moreover, $+1$ in the left formula has little impact on the inequation. Therefore, from the previous formula we can infer that

$$\frac{\$\left(\frac{\mu_k}{|X_k - X_j|^\alpha}\right)}{\$\left(\frac{\mu_i}{|X_i - X_j|^\alpha}\right)} \leq \frac{\$}{\$ + 1},\tag{19}$$

where $\alpha$ is the distance decay and $\mu$ replaces entanglement $\lambda$ in (3) to avoid conflicts with the throughput $\lambda$. After expanding $\$(\mu)$, we have

$$\frac{|X_i - X_j|^\alpha}{|X_k - X_j|^\alpha} \frac{\mu_k}{\mu_i} \leq \frac{\$}{\$ + 1} . \tag{20}$$

In the worst situation, $\frac{\mu_k}{\mu_i}$ is equal to $\frac{\mu_\top}{\mu_\perp}$ where $\mu_\top$ is the maximum feasible entanglement in the scene and $\mu_\perp$ is the minimum feasible entanglement to lead to the emergence of communications in the scene which meets that $\forall \mu > \mu_\perp$ implies $q^*(\mu) > 0$ and $\$^*(\mu) > 0$. Then, we have the inequation with the distance and incentive schemes:

$$|X_k - X_j| \geq \left( \frac{\mu_\top}{\mu_\perp} \frac{\$ + 1}{\$} \right)^{\frac{1}{\alpha}} |X_i - X_j|$$
$$= (1 + \Delta)|X_i - X_j| , \tag{21}$$

where $\Delta := \left( \frac{\beta P_{\min}}{P_{\max}} \right)^{\frac{1}{\alpha}} - 1$.

*4.5. Upper Bound of Network Capacity*

Different from the other studies, because $\Delta$ is deemed as a variable associated with incentive schemes by (21), it is not simplified during derivations. According to the interference restriction and the meeting probability among nodes and clusters, the amount of messages that can be transmitted in a time slot between clusters is $O\left( \frac{mR^2}{\Delta^2} \right)$ if the transmission range is $\Theta\left( R\sqrt{\frac{m}{n}} \right)$ and the amount of messages in a cluster is the similar. Therefore, we can propose the following theorem:

**Theorem 1** (Throughput capacity of incentive schemes)**.** *In a sparse clustering regime with incentive schemes, the throughput capacity can achieve $\Theta\left( \frac{mR^2}{\Delta^2} \right)$ and the upper bound can be reached if and only if the transmission range satisfies $l_i = \Theta\left( R\sqrt{\frac{m}{n}} \right)$, where i is a duplication in a cluster, and $l_i = \Theta(1)$, where i is a duplication between clusters. Each node can achieve $\Theta\left( \frac{mR^2}{n\Delta^2} \right)$.*

We dismiss the proof here for simplification, as it is similar to the proofs of Theorem 1 and Theorem 2 in [19].

The above conclusion does not consider the limited ratio resources that messages cannot be sent out due to congested channels when many nodes send messages simultaneously in the same area. If the channel capacity can transmit $W$ bits per second during the whole network alive time $T$, each bit $b$ at $h$th-hop satisfies

$$\sum_{b=1}^{\lambda nT} \sum_{h=1}^{h(b)} 1 \leq \frac{WnT}{2} \leq WnT . \tag{22}$$

According to (16), there are no other nodes in the disk radius $\frac{\Delta}{2}$ times transmission range. Each node consumes $\frac{1}{n}$ in area $n$ averagely, thus, the summing area that source nodes consumes meets

$$\sum_{b=1}^{\lambda nT} \sum_{h=1}^{h(b)} S_b \leq WT , \tag{23}$$

where $S_b$ is the area of bit $b$ and $h(b)$ is the hop of bit $b$. The following lemma can be achieved with the above idea:

**Lemma 1** (Throughput inequation considering ratio resources). *Considering ratio resources, the throughput of bit b under sparse cluster regime should meet the following inequation:*

$$\sum_{b=1}^{\lambda^s nT} \frac{\Delta^2}{4} \frac{\mathbb{E}[R_{d_b^s}] - 1}{n} + \mathbb{E}\left[\sum_{b=1}^{\lambda^s nT} \sum_{h=1}^{h_b^s + \frac{nRc_b^s}{mR^2}} \frac{\pi \Delta^2}{4} \frac{l_b^{h2}}{mR^2}\right] \le c_1^s WT \log n\,, \tag{24}$$

*where $c_1^s$ is a positive constant.*

The proof is omitted for simplification as it is similar to Proposition 3 in [20] and Lemma 3.3 in [2]. With the above lemma and theorem, we can deduce the following theorem:

**Theorem 2** (throughput capacity under sparse clustering regime with incentive schemes). *In a sparse clustering regime, the throughput capacity with incentive schemes is*

$$O\left(\min\left\{\sqrt{\frac{n \log n}{mR^2\Delta^2}}, \frac{mR^2}{n\Delta^2}\right\}\right) \tag{25}$$

*and the upper bound can be achieved when*

$$l = \Theta\left(R\sqrt{\frac{m}{n}}\right).$$

The proof is shown in Appendix A.

## 5. Factors Discussion

To design efficient incentive schemes with the formula, it is necessary to take a detailed discussion for each factor. The cooperation degree is introduced as an important factor for the incentive degree that provides a base of a network, as discussed in Section 2.1, so we discuss it first. Then, the mobility mode is analyzed because the capacity formula without incentive schemes should have the same result as the sparse clustering regime. Finally, each factor in utility $\$^*$ reveals impacts.

### 5.1. Cooperation Degree

As $\mu_\top$ and $\mu_\perp$ in (21) are defined by the network environment, intuitively, Theorem 1 should meet $\lim_{\Delta \to \infty} \lambda^s = 0$ and $\lim_{\Delta \to 1} \lambda^s = O\left(\frac{mR^2}{n}\right)$. The first proposition is easily derived because $\lim_{\mu \to \infty} \$ = \infty$ theoretically. When $\lim_{\$ \to \infty} \Delta = \left(\frac{\mu_\top}{\mu_\perp}\right)^{\frac{1}{\alpha}} - 1 = 1$, the second proposition is derived. It must be noted that $\mu$ has explicit upper and lower bounds in reality because nodes cannot communicate to others without restriction. Therefore, the real upper bound should consider the cooperation between nodes defined by network topology and mobility.

### 5.2. Mobility Mode

Without incentive schemes, $m$, $R$, and $n$, represented as mobility mode, are the main factors. The capacity defined in Theorem 2 is a minmum value from two formulas. To find the impactful range of each formula, the variation of network capacity versus the number of nodes is shown with results in Figure 6. With an increasing number of nodes, network capacity grows quickly at the beginning. It converges at last due to the limited ratio resources. The red vertical line is the break point represented as the ideal number of nodes in a special network. Here, $m$ and $R$ have corresponding results as above because they are inversely proportional to $n$.

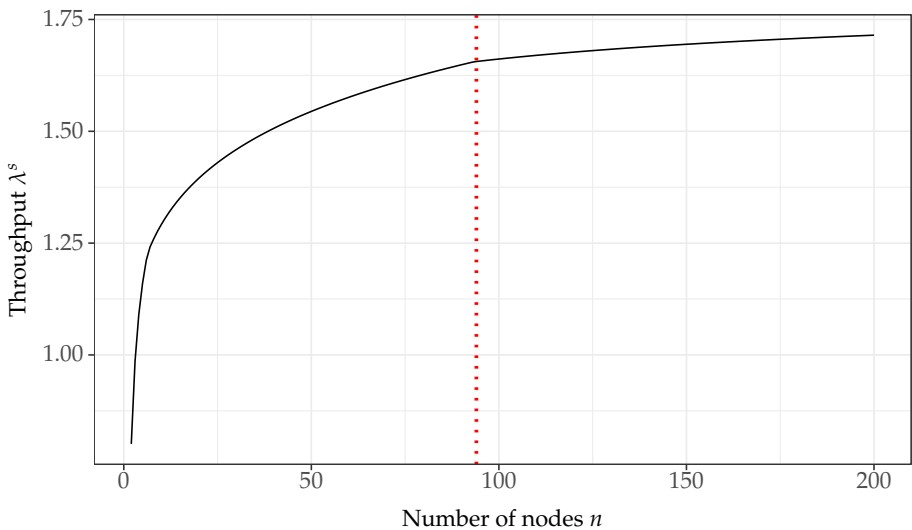

**Figure 6.** Number of nodes versus throughput.

*5.3. Incentive Factors*

In the following discussion under the two different formulas, the part from Theorem 1 in Theorem 2 is discarded for simplicity. The parameters used in analysis are listed in Table 3, and are selected according to simulations in Section 3 and cluster sparse regime [2].

**Table 3.** Value of parameters.

| Parameter | Description | Value |
|---|---|---|
| $n$ | number of node | 100 |
| $v$ | Grow exponent of $m$ | 4/9 |
| $\beta$ | Growth exponent of $R$ | 1/3 |
| $\mu_\top$ | Maximum achievable entanglement | 10 |
| $\mu_\perp$ | Minimum achievable entanglement | 5 |
| $\alpha$ | Decreasing coefficient of cooperation | 1 |
| $\mu$ | Entanglement | 6 |
| $c$ | Constant cost | 0 |
| $s$ | Safty coefficient | 2 |
| $E_c$ | Energy cost per message | 1 |
| $B_c$ | Buffer cost per message | 1 |
| $E_s$ | Energy capacity | 5 |
| $B_s$ | Message buffer | 5 |
| $Z$ | Message price | 100 |
| $R$ | Message reward | 30 |

How incentive schemes affect network capacity is shown in Figure 7. An incentive scheme usually tries its best to increase its incentive degree to reach a higher capacity. However, the decreasing rate of change suggests that throughput cannot increase infinitely. The limitation is restricted by the cooperation degree discussed in Section 5.1. Moreover, the incentive degree has its own extremum, as shown in Section 3. Therefore, thoughput can reach its limitation earlier.

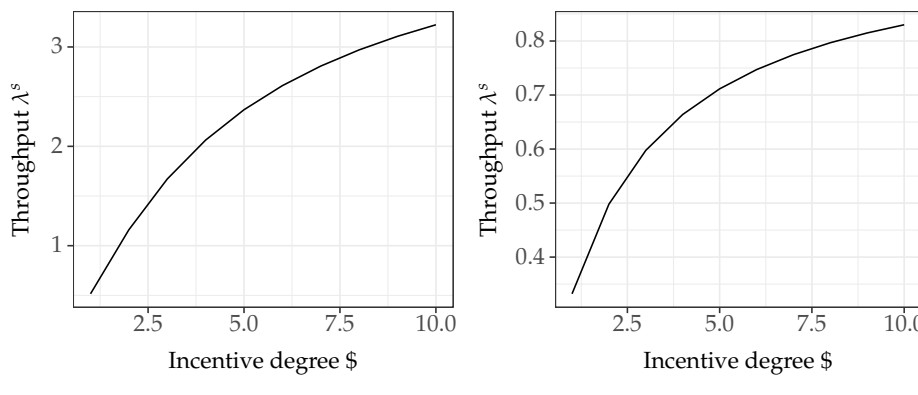

(**a**) Without ratio resources　　　　　　　　(**b**) With ratio resources

**Figure 7.** Incentive schemes versus throughput.

The limitation can be observed obviously through the relation with buffer usage and energy usage. Similar to the analysis in Section 3, the message size and energy cost per message work with buffer size and energy capacity together. As such, only the latter are discussed here; the relations are shown in Figures 8 and 9. The fact that they reach the limitation quickly suggests that both of them are the basic condition to ensure that nodes communicate. If a higher network capacity is needed, other factors should be adjusted to increase the achievable upper bound. According to this analysis, a suitable buffer and energy are more important; their value can be taken from our theorem.

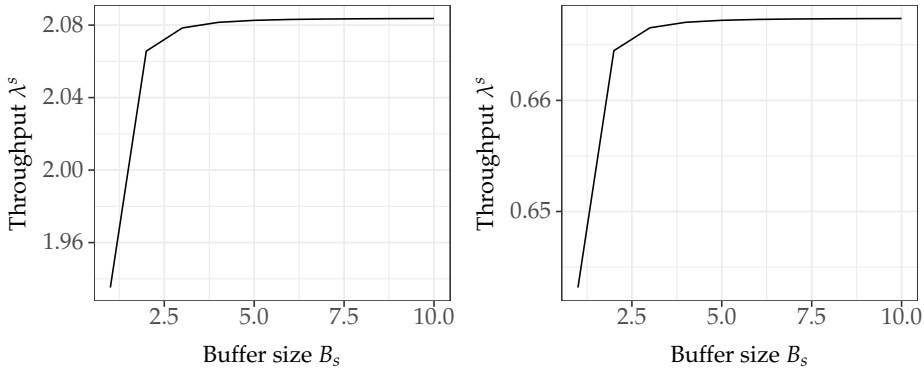

(**a**) Without ratio resources　　　　　　　　(**b**) With ratio resources

**Figure 8.** Buffer size schemes versus throughput.

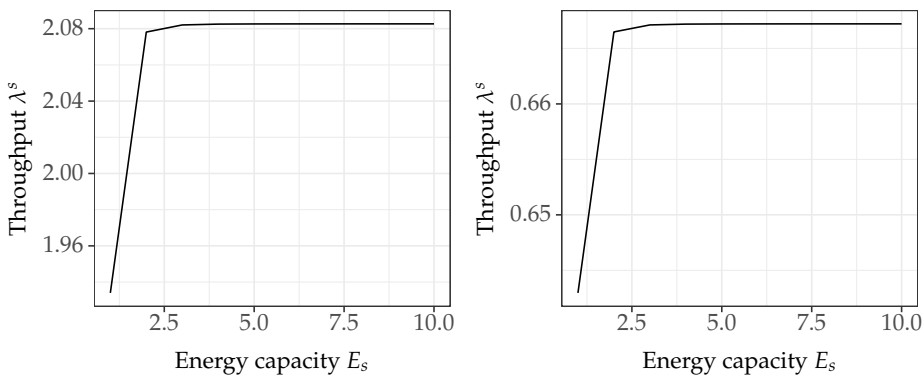

(**a**) Without ratio resources　　　　　　　　(**b**) With ratio resources

**Figure 9.** Energy capacity versus throughput.

The secure and incentive scheme can affect the achievable upper bound shown in Figures 10 and 11. Both of them have negative effects on network capacity. However, we cannot reduce them discretionarily. A too low reward or a too high price may block communications between nodes due to insufficient credits. A too low security coefficient may bring attacks to nodes. With these conditions, the relations can help us to design incentive schemes to select rewards and evaluate the security coefficient for both a node and the network.

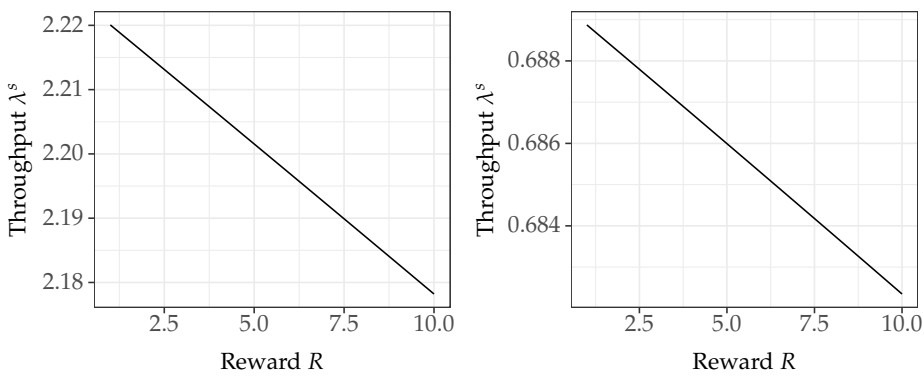

(**a**) Without ratio resources        (**b**) With ratio resources

**Figure 10.** Reward versus throughput.

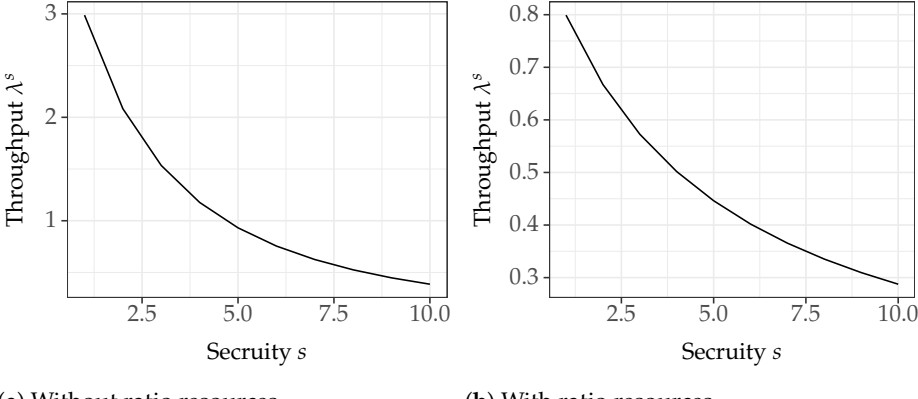

(**a**) Without ratio resources        (**b**) With ratio resources

**Figure 11.** Security versus throughput.

## 6. Conclusions

In this paper, we design a mathematical model based on a quantum game model involving node parameters and use this model with a sparse clustering regime to analyze network capacity with incentive schemes. The mathematical model reflects the incentive degree for an incentive scheme in opportunistic networks with selfish nodes, finding that the character of incentive schemes is mainly determined by the relationship between reward and cost. According to the capacity analysis, how the factors affects on capacity are aware. The formula can be a guideline for new efficient incentive schemes. However, only energy usage, buffer usage, and security are considered. Other parameters, such as messages alive time and how the environment affects entanglement, can be taken into account as well. The capacity analysis associates the transmission model with incentive schemes simply. In the future, a more powerful model should be raised to reveal more details. In addition to sparse clustering regimes, other mobility models can be analyzed with incentive schemes.

**Author Contributions:** Conceptualization, S.J. and R.F.; methodology, R.F.; software, R.F.; validation, R.F.; formal analysis, R.F.; investigation, R.F.; resources, R.F. and S.J.; data curation, R.F.; writing—original draft preparation, R.F.; writing—review and editing, S.J. and Z.Z.; visualization,

R.F.; supervision, Shengming Jiang; project administration, S.J.; funding acquisition, S.J. All authors have read and agreed to the published version of the manuscript.

**Funding:** This research recevied was funded by The Innovation Program of Shanghai Municipal Education Commission of China, grant number 2021-01-07-00-10-E00121.

**Institutional Review Board Statement:** Not applicable.

**Informed Consent Statement:** Not applicable.

**Data Availability Statement:** Not applicable.

**Conflicts of Interest:** The authors declare no conflict of interest.

**Appendix A. Proof of Theorem 2**

Applying Cauchy–Schwarz inequality to the second monomial in (24) obtains

$$
\left( \sum_{b=1}^{\lambda^s nT} \sum_{h=1}^{h_b^s + \frac{nRc_b^s}{mR^2}} l_b^h \right)^2 \le \left( \sum_{b=1}^{\lambda^s nT} \sum_{h=1}^{h_b^s + \frac{nRc_b^s}{mR^2}} 1 \right) \left( \sum_{b=1}^{\lambda^s nT} \sum_{h=1}^{h_b^s + \frac{nRc_b^s}{mR^2}} l_b^h \right)
$$

$$
\le \frac{WTn}{2} \sum_{b=1}^{\lambda^s nT} \sum_{h=1}^{h_b^s + \frac{nRc_b^s}{mR^2}} (l_b^h)^2 ,
$$

where the last step is from (22). Then, we have

$$
\mathbb{E}\left[ \sum_{b=1}^{\lambda^s nT} \sum_{h=1}^{h_b^s + \frac{nRc_b^s}{mR^2}} (l_b^h)^2 \right] \ge \frac{2}{WTn} \mathbb{E}\left[ \left( \sum_{b=1}^{\lambda^s nT} \sum_{h=1}^{h_b^s + \frac{nRc_b^s}{mR^2}} (l_b^h) \right)^2 \right].
$$

Applying Jensen inequality to the above inequation obtains

$$
\frac{2}{WTn} \mathbb{E}\left[ \left( \sum_{b=1}^{\lambda^s nT} \sum_{h=1}^{h_b^s + \frac{nRc_b^s}{mR^2}} (l_b^h) \right)^2 \right] \ge \frac{2}{WTn} \left( \mathbb{E}\left[ \sum_{b=1}^{\lambda^s nT} \sum_{h=1}^{h_b^s + \frac{nRc_b^s}{mR^2}} (l_b^h) \right] \right)^2 ,
$$

where $l_b^s$ is the transimssion range of bit $b$ from source $s$ and $r_b^s$ is the distance of each transmission between two nodes. For multi-hops, the following inequation is obvious:

$$
\sum_{h=1}^{h_b^s} l_b^h \ge l_b^s .
$$

Appling Jensen inequality yields

$$
\frac{2}{WTn} \left( \mathbb{E}\left[ \sum_{b=1}^{\lambda^s nT} \sum_{h=1}^{h_b^s + \frac{nRc_b^s}{mR^2}} (l_b^h) \right] \right)^2 \ge \frac{2}{WTn} \left( \sum_{b=1}^{\lambda^s nT} \mathbb{E}[l_b^s] \right)^2
$$

$$
\mathbb{E}\left[ \sum_{b=1}^{\lambda^s nT} \sum_{h=1}^{h_b^s + \frac{nRc_b^s}{mR^2}} (l_b^h)^2 \right] \ge \frac{2}{WTn} \left( \sum_{b=1}^{\lambda^s nT} \mathbb{E}[l_b^s] \right)^2 .
$$

Substituting the above formula into (24) yields

$$\sum_{b=1}^{\lambda^s nT} \frac{\mathbb{E}[R_{d_b}^s] - 1}{n} + \frac{2\pi}{WTnmR^2} \left(\sum_{b=1}^{\lambda^s nT} \mathbb{E}[l_b^s]\right)^2 \leq \frac{4c_1^s WT \log n}{\Delta^2}. \tag{A1}$$

To continue the calculation, we should discuss $R_{d_b^s}$. Let $l_b^s\hat{}(t)$ be the minimum distance between the current cluster to $C_d$ in slot $t$. If the value is negative, there is an overlap between the current cluster and $C_d$. For convenience, we define $L_b^s\hat{}(t) = \max(0, l_b^s\hat{}(t))$. For bit $b$, considering the network area and the transmission area of a cluster yields

$$\mathbb{E}\left[\frac{n}{[R + L_b^s\hat{}(t)]^2}\right] = \mathbb{E}\left[\frac{n}{R^2}\mathbb{I}_{L_b^s(t) \leq 0}\right] + \mathbb{E}\left[\frac{n}{[R + l_b^s\hat{}(t)]^2}\mathbb{I}_{L_b^s(t) > 0}\right]. \tag{A2}$$

For the second monomial, we can obtain

$$\begin{aligned} \mathbb{E}\left[\frac{n}{[R + l_b^s\hat{}(t)]^2}\mathbb{I}_{L_b^s(t) > 0}\right] &= \int_0^{\sqrt{n}} \frac{n}{(R + u)^2} d\Pr\left[l_b^s\hat{}(t) \leq u\right] \\ &= 1 - \frac{n}{R^2}\Pr\left[l_b^s\hat{}(t) \leq 0\right] \\ &\quad + \int_0^{\sqrt{n}} \frac{2n}{(R + u)^3}\Pr\left[l_b^s\hat{}(t) \leq u\right] du. \end{aligned} \tag{A3}$$

Therefore, the original inequation is

$$\begin{aligned} \mathbb{E}\left[\frac{n}{[R + L_b^s\hat{}(t)]^2}\right] &= 1 + \int_R^{\sqrt{n}} \frac{2n}{u'^3}\Pr\left[R + l_b^s\hat{}(t) \leq u\right] du' \\ &= 1 + \int_R^{\sqrt{n}} 2\pi R_{c_b}^s(t)\frac{(R + u')^2}{u'^3} du' \\ &\leq 1 + 6\pi R_{c_b}^s(t)\int_R^{\sqrt{n}} \frac{1}{u'} du \\ &= 1 + 6\pi R_{c_b}^s(t)\log\frac{\sqrt{n}}{R} \\ &\leq 1 + 6\pi R_{c_b}^s(t)\log n \\ &\leq 6\pi R_{c_b}^s(t)\log n. \end{aligned} \tag{A4}$$

For a message under all slots, we have

$$\begin{aligned} \mathbb{E}[(R + \hat{L}_b^s)^2] &\leq (R + l_b^s)^2 \\ \mathbb{E}\left[\frac{n}{[R + \hat{L}_b^s]^2}\right] &\geq \frac{n}{(R + l_b^s)^2} \\ \frac{n}{(R + l_b^s)^2} &\leq 6\pi R_{c_b}^s \log n \\ R_{c_b}^s &\geq \frac{n}{6\pi \log n(R + l_b^s)^2} \\ &\geq \frac{n}{24\pi \log nRl_b^s}. \end{aligned} \tag{A5}$$

Substituting the above inequation into (A1) yields

$$\frac{1}{24\pi \log nR\sum_{b=1}^{\lambda^s nT} \mathbb{E}[l_b^s]} + \frac{2\pi}{WTnmR^2}\left(\sum_{b=1}^{\lambda^s nT} \mathbb{E}[l_b^s]\right)^2 \leq \frac{4c_1^s WT \log n}{\Delta^2}. \tag{A6}$$

If $h_b^s < \frac{nR_{c_b^s}}{mR^2}$, the ratio resources consumption does not decrease with a decreasing number of hops due to the unchanging transmission range. In such a situation, we deem $h_b^s = \Theta(\frac{nR_{c_b^s}}{mR^2})$, which is contained in the other situation and has the same result.

If $h_b^s \geq \frac{nR_{c_b^s}}{mR^2}$, we can calculate (A6) directly. If $\sum_{b=1}^{\lambda^s nT} \mathbb{E}[l_b^s] < \lambda^s TmR^2$, the first monomial has little impact on the result in (A6), and it is duplication in clusters which is restricted by duplication in clusters according to Theorem 1. We ignore this term and obtain

$$
\begin{aligned}
\frac{2\pi}{WTnmR^2}\left(\lambda^s TmR^2\right)^2 &\leq \frac{4c_1^s WT \log n}{\Delta^2} \\
\lambda^{s2} &\leq \frac{c_1^s nW^2 \log n}{mR^2\Delta^2} \\
\lambda^s &\leq O\left(\sqrt{\frac{n \log n}{mR^2\Delta^2}}\right)
\end{aligned}
\tag{A7}
$$

If $\sum_{b=1}^{\lambda^s nT} \mathbb{E}[l_b^s] \geq \lambda^s TmR^2$, according to Theorem 1, the same upper bound can be achieved when $l_i = \Theta\left(R\sqrt{\frac{m}{n}}\right)$.

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
