# Peer review of "Capacity Analysis of Incentive Schemes in Opportunistic Networks"

_jmse, doi:10.3390/jmse10101474_

Round 1

Reviewer 1 Report (Previous Reviewer 1)

The topic presented in this article is interesting. However, the paper presentation is bit confusing for the readers.

Also, the work presented in this article is not fitting to the scope of the journal and special issue.

Following changes are required to be addressed:

Equation Numbering is missing in most of the paper.
Abbreviations like TCP, P2P are used in the paper without actually defining them.

Reviewer 2 Report (Previous Reviewer 2)

Dear Authors

Now, It is much better.

Author Response

Thank you very much for your time involved in reviewing the manuscript and your
very encouraging comments on the merits.

Reviewer 3 Report (New Reviewer)

The authors present network capacity analysis based on the network modeling from quantum game theory. The technical part is sounded. However, the modeling part is not quite clear. The authors may want to improve the presentation and connect the quantum game theory to the content. For example, what does the entanglement mean in the opportunistic networks? Why quantum game theory is needed? Why quantum state modeling is better than the canonical state definition? 

Author Response

We apply your advice to express our model more detail. In Section 1, we add why quantum game theory is better than the cononical one
and why it is applied. In Section 2, we add more explanation on quantum game model such as the three stages of the model, parameters meanings and the connect between the model and opportunistic networks.

What does the entanglement mean in the opportunistic networks?

(In Section 2)
The entanglement means the cooperation degree determined by the scenario such as selfish degrees, social groups, movement paths and etc.

Why quantum game theory is needed and why quantum state modeling is better than the canonical state definition?

(In Section 1)
Game theory is used to state nodes' strategies to analyze network traffic in TCP
with selfish nodes and P2P networks with free-riding problem, and some analyatic models for different categories based on classical game theory is proposed which don't find common characters of incentive schemes. A generic model is built with quantum game theory because quantum game theory extends the strategy space to search optimal strategies from a wider range and the concept of entanglement to depict the complex interaction among nodes accurately in the decision-making process. A model with these features can evaluate most of incentive schemes
and reveal the common character.

We would like to take this opportunity to thank you for all your time involved
and this great opportunity for us to improve the manuscript. We hope you will
find this revised version satisfactory.

Sincerely,

Ruoyu Feng

Round 2

Reviewer 3 Report (New Reviewer)

The authors improved the manuscript. All of concerns have been addressed. 

This manuscript is a resubmission of an earlier submission. The following is a list of the peer review reports and author responses from that submission.

Round 1

Reviewer 1 Report

The work is suitable for getting published in this journal

Reviewer 2 Report

Dear Authors

 Your paper is very interesting, especially because it deals with opportunistic networks, you propose such a model to show the incentive degree with the incentive scheme, cooperation degree, energy usage, buffer usage, and security based on a quantum game model.

 1. In opportunistic networks a very important variable is the contact-time between nodes, it would be good to analyze your results based on the contact time of the nodes.

2. In the parameters of the simulations, you propose a transmission range of 300m, based on what criteria do you define this value?

3. Have you considered the effects of the environment or scenario of the nodes?

 4. Explain more about the type of mobility of the nodes.

5. Explain more about the latency in message delivery with and without applying your collaboration proposal.

 6. Also, it would be great if you could use real mobility traces in your simulations, there are many.

Reviewer 3 Report

The objective and research questions of this manuscript are unclear, same as its importance to the wide field of scientific community;

Although many models of networks are discussed, their applicability to real-life problems is not even mentioned once throughout the paper. This makes a reader confused of what kinds of networks are referred to;

With no reference to any of Subject Areas being of interest to JMSE, I must say that the presented manuscript falls outside of the scope of the journal and must therefore be rejected.